# Sensing Cells-Peptide Hydrogel Interaction In Situ via Scanning Ion Conductance Microscopy

**DOI:** 10.3390/cells11244137

**Published:** 2022-12-19

**Authors:** Tatiana N. Tikhonova, Vasilii S. Kolmogorov, Roman V. Timoshenko, Alexander N. Vaneev, Dana Cohen-Gerassi, Liubov A. Osminkina, Petr V. Gorelkin, Alexander S. Erofeev, Nikolay N. Sysoev, Lihi Adler-Abramovich, Evgeny A. Shirshin

**Affiliations:** 1Department of Physics, M.V. Lomonosov Moscow State University, Leninskie Gory 1/2, 119991 Moscow, Russia; 2Laboratory of Biophysics, National University of Science and Technology “MISiS”, 4 Leninskiy Prospekt, 119049 Moscow, Russia; 3Department of Chemistry, M.V. Lomonosov Moscow State University, Leninskie Gory 1/3, 119991 Moscow, Russia; 4Department of Oral Biology, The Goldschleger School of Dental Medicine, Sackler Faculty of Medicine, The Center for Nanoscience and Nanotechnology, The Center for the Physics and Chemistry of Living Systems, Tel Aviv University, Tel Aviv 69978, Israel; 5World-Class Research Center “Digital Biodesign and Personalized Healthcare”, Sechenov First Moscow State Medical University, 8-2 Trubetskaya St., 119991 Moscow, Russia

**Keywords:** scanning ion conductance microscopy, hydrogel, peptide self-assembly, cells, fibrillation, regenerative medicine, reactive oxygen species

## Abstract

Peptide-based hydrogels were shown to serve as good matrices for 3D cell culture and to be applied in the field of regenerative medicine. The study of the cell-matrix interaction is important for the understanding of cell attachment, proliferation, and migration, as well as for the improvement of the matrix. Here, we used scanning ion conductance microscopy (SICM) to study the growth of cells on self-assembled peptide-based hydrogels. The hydrogel surface topography, which changes during its formation in an aqueous solution, were studied at nanoscale resolution and compared with fluorescence lifetime imaging microscopy (FLIM). Moreover, SICM demonstrated the ability to map living cells inside the hydrogel. A zwitterionic label-free pH nanoprobe with a sensitivity > 0.01 units was applied for the investigation of pH mapping in the hydrogel to estimate the hydrogel applicability for cell growth. The SICM technique that was applied here to evaluate the cell growth on the peptide-based hydrogel can be used as a tool to study functional living cells.

## 1. Introduction

Microscopy-based methods have been widely utilized for cell and tissue imaging and assessing various cellular processes, such as cell-cell interactions, cell signaling, and differentiation [1,2,3,4]. Specifically, these methods include fluorescence microscopy, which is commonly-used due to its non-invasiveness, as well as spatial and temporal resolution [5,6,7,8], and atomic force microscopy (AFM), which can facilitate imaging of individual molecules in a native-like environment with a sub-nanometer resolution. Yet, these methods bear certain disadvantages, since fluorescence imaging mainly requires the use of specific exogenous probes, which may potentially affect the function of certain cell mechanisms, and AFM probing has a number of limitations for long-term measurements [9].

Scanning ion-conductance microscopy (SICM) is a powerful imaging technique for the non-invasive topographical mapping of complex biological samples, e.g., cancer cells, neurons, cardiomyocytes, etc., with a nanoscale resolution in a physiological environment [10,11,12,13]. SICM also allows the mapping of the cell stiffness in a low-stress mode at a lateral and vertical resolution better than 100 nm, and monitoring the dynamic changes of the cell membrane induced by various external stimuli. In this case, much lower forces can be applied for the study of nanomechanical properties compared to AFM [14]. Moreover, SICM allows the assessment of redox processes in living cells by applying a special nanopipette with a nanoelectrode sensor [15], recording ion currents and cell membrane potentials using the patch-clamp method [16], and the label-free pH monitoring of living cancer cells [17,18].

In SICM, the scanning probe is an electrolyte-filled glass pipette with a radius from several to hundreds of nanometers. The main Ag/AgCl electrode is placed inside the pipette while the reference Ag/AgCl electrode is in the Petri dish with the sample. The potential bias is applied between two electrodes, which ensures the flow of the ion current through the tip of the nanocapillary. As the nanopipette tip approaches the substrate surface, the current begins to decrease and drops substantially when the probe–sample distance is smaller than the probe tip radius. To perform topographical sample mapping, a feedback system is used based on the registration of the ion current flowing through the pipette and the coordinates of the piezo actuators. As a result, the comparison of the scanning probe positions during the entire scan gives a detailed topographical map that is reconstructed in a noncontact manner [19].

In this work, we focused on the analysis of SICM application in studying cells’ interaction with biomaterials through a model system of a peptide-based hydrogel. Injectable hydrogels possess important properties such as porous structure for cell proliferation and transplantation, similarity to the natural extracellular matrix, etc., for the application as 3D cell culture scaffolds in tissue engineering [20,21,22]. A variety of biomaterials have been used for the preparation of injectable hydrogels such as collagen, chitosan, alginate, heparin, etc. [23,24,25,26]. Aromatic moieties such as fluorenylmethoxycarbonyl (Fmoc) self-assembly into a fibrillar hydrogel in aqueous solutions, as the aromatic group mediates both π–π stacking and hydrophobic interactions [27]. The Chinese hamster ovarian cells, fibroblasts, and preosteoclasts, were used to demonstrate the potential employment of modified self-assembled peptide hydrogels as scaffolds for tissue engineering due to their biocompatibility and good adhesion properties [28,29,30,31].

Here, the surface topography of a hydrogel self-assembled by the N-fluorenylmethyloxycarbonyl-diphenylalanine (Fmoc-FF) peptide in an aqueous solution was assessed by SICM during the formation process. Moreover, the interaction between cells and the Fmoc-FF peptide-based hydrogel was studied using SICM, aimed at the assessment of SICM capabilities in measuring cell response resulting from the biomechanical and biochemical properties of the hydrogel. Also, a zwitterionic label-free pH nanoprobe was used for the characterization of hydrogel properties that could have an effect on cell viability. The SICM technique was applied to measure the reactive oxygen species (ROS) content in cells grown on the Fmoc-FF hydrogel as a marker of cell-hydrogel interaction and its influence on the cell biochemistry. The obtained results of the in situ assessment of peptide self-assembly and cell-hydrogel interaction pave the way for the precise and minimally invasive testing of biomaterials and their biocompatibility for applications in regenerative medicine.

## 2. Materials and Methods

### 2.1. Materials

(1)Reagents. All chemical reagents used were of analytical grade and all aqueous solutions were prepared with ultrapure water, with resistivity not less than 18 MΩcm. The medium was purchased from Thermo Fisher Scientific (Walthman, MA, USA). The HBSS buffer solution was purchased from PanEco, Russia. Phosphate buffered saline (PBS, 10 mM Na_2_HPO_4_, 2.7 mM KCl and 137 mM NaCl, pH = 7.4) was prepared by dissolving tablets (Sigma, Saint Louis, MO, USA) in 200 mL deionized water (Milli-Q; Millipore Corp, Burlington, MA, USA). Poly-l-lysine (PLL, P4707), glucose oxidase (GOx, G2133), NaOH, HCl and glutaraldehyde were purchased from Sigma–Aldrich.(2)Hydrogel. The Fmoc-FF was obtained from GL Biochem (Shanghai, China). Thioflavin T (ThT) was purchased from Sigma-Aldrich (Berlin, Germany). The solvent-switch method was used for the preparation of Fmoc-FF hydrogel: Fmoc-FF stock solution that was initially dissolved in dimethyl sulfoxide (DMSO) (100 mg/mL) was diluted with double distilled water to obtain the final concentration 5 mg/mL, at that the final DMSO concentration was 5%. It is known that 5% DMCO concentration in hydrogel can decrease cell survival that are placed inside hydrogel [32]. To avoid this effect, the hydrogel was washed for 1 day with PBS buffer to remove the excess of DMSO [28]. Due to this procedure, firstly, the turbid Fmoc-FF solution was observed that became transparent with time, when the hydrogel is formed.(3)Cell culture. (Bacя) MCF-7 (human breast carcinoma) (ATCC, Manassas, VA, USA) cells were cultured in DMEM/F12 (Gibco, Saint Louis, MO, USA) culture medium with 10% fetal bovine serum (Gibco, USA), 50 u/mL penicillin, 0.05 mg/mL streptomycin (Gibco), and 1 × S3 GlutaMAX (Gibco) in cell culture flask T-75. Then, cells were resuspended with DPBS (Gibco, USA) and Trypsin LE (Gibco, USA) and cultured in 35-mm Petri plastic dish or glass bottom with hydrogel. After 24 h incubation in DMEM/F12 (Gibco) culture medium in 5% CO_2_ and 37C. Before scanning procedure or ROS measurements cells were gently washed by Hank’s solution (Gibco, USA).

### 2.2. Scanning Ion Conductance Microscopy (SICM) Measurements

Non-contact topography mapping was made using SICM by ICAPPIC (ICAPPIC Limited, London, UK). Nanopipettes were made using a P-2000 laser puller (Sutter Instruments, Novato, CA, USA) from borosilicate glass capillaries (O.D. 1.2 mm, I.D. 0.90 mm, 7.5 cm length). Typical inner radius (r) of the nanopipettes was in the range of 40–50 nm for all experiments with hydrogel and living cells. Hydrogel and cell topography were performed using the noncontact hopping mode with a scan size 40 × 40 µm, with the final images resolution to 256 × 256 pixels. The fall rate during imaging was set at 120 µm s^−1^. The measurements were carried out at 20 °C.

### 2.3. Fluorescence Lifetime Imaging Microscopy (FLIM) Measurements

For Fluorescence lifetime imaging microscopy (FLIM) measurements MicroTime 200 STED microscope (PicoQuant GmBH, Berlin, Germany) was used, the excitation source was 405 nm laser. The pulse duration was 40 ps, pulse rate was 40 MHz, maximum power was 50 μW. The immersion objective ((UplanSApo, Olympus, Japan) with a 100 × 1.4 NA was applied. To detect the fluorescence emission the single photon-counting modules (Excelitas, New York, NY, USA) were used. The studies were carried out at 20 °C. For FLIM measurements. The fluorescence dye Thioflavin T was used with the concentration 40 µM.

### 2.4. Confocal Microscopy Measurements

For confocal microscopy measurements, all the samples were placed in Petri dishes with glass bottom and analyzed using a confocal laser scanning microscope LSM 880. The 40× water immersion objective, AiryScan module and GaAsP detector and were used. Excitation was performed at a wavelength of 405 nm. The measurements were carried out at 20 °C.

### 2.5. Label-Free pH Nanoprobe Measurements

Preparation of pH-sensitive nanomembrane probes has been described in detail previously [17]. Commercially available quartz glass capillary QF120-60-7.5 (O.D. 1.2 mm, I.D. 0.6 mm, Sutter Instruments Co., Novato, CA, USA) were used to fabricate nanopipettes. Nanopipettes were pulled using laser-based P-2000 pipette puller (Sutter Instruments Co., USA) with a one-step protocol containing the following parameters: heat 560, filament 2, velocity 25, delay 145 and pull 180. The pulled quartz nanopipette was filled to a length of about 1 mm with a mixture of PLL and GOx (GOx was inactivated by denaturation at 70 °C for 10 min and then dissolved in 0.01% (*v*/*v*) PLL at a concentration of 0.4 mg/mL) by capillary action. To construct the nanomembrane at the tip of the pipette the crosslinking reaction was performed between GOx and PLL, this tip was left in the vapors of 25% (*v*/*v*) glutaraldehyde for overnight. Then, the nanomembranes were washed with 100 mM  KCl to get rid of un-crosslinked chemicals.

The setup for voltametric experiments consisted of patch-clamp amplifier MultiClamp 700B (Axon Instruments, Whipple, CA, USA) and ADC-DAC converter Axon Digidata 1550B (Axon Instruments, Burlingame, CA, USA). The amplifier head was fixed on a PatchStar Micromanipulator (Scientifica, London, UK). In the experiment the inverted optical microscope Mikromed-I-LUM (Mikromed, Russia) was used. A pH-sensitive nanopipette was fixed on the head of the amplifier with the help of special holder. The potential difference between the pH-sensitive nanopipette and the reference electrode (Ag/AgCl) was obtained with the help of the pClamp 11 software suite (Molecular Devices, Silicon Valley, CA, USA).

Before and after experiment, each pH-sensitive nanoprobe was calibrated with the use of standard pH buffer with different pH values (4–9). To investigate the pH level, the potential was cycled between −0.6 V and +0.6 V vs. Ag/AgCl at a scan rate of 650 mV s^−1^. Calibration curve obtained by current measurement at +0.6 V.

Samples were incubated for 16 h with PBS buffer (pH same as gel). The measurements were carried out without changing the buffer inside and outside the gel with a step of 20 µm. Each point was recorded twice. pH levels were determined based on the calibration curve. Results are shown as mean (average over several technical repetitions) ± SEM. The measurements were carried out at 20 °C.

### 2.6. Reactive Oxygen Species Measurements

The measurements were performed using the setup described in the section Label-free pH nanoprobe measurements. Cells adhered to the surface of the Petri dish and to the hydrogel were thoroughly washed with a buffer solution HBSS. Next, the Petri dish was placed on the object stage of an optical microscope. Cells well attached to the surface were selected for further experiments. After that, the nanoelectrode was positioned using a Scientifica manipulator. For more accurate positioning of the nanoelectrode to the cells, software was used that allows moving the nanoelectrode in strictly specified steps (LinLab2). The current in the system increased when the electrode penetrated the cell, after that a slow decrease occurred. Then, the electrode was removed from the cell, the current quickly returned to its initial level. To assess the ROS generation, the difference between the currents inside the cell and outside, after removing the electrode from the cell, was used. The measurements were carried out at 20 °C.

## 3. Results and Discussion

### 3.1. Peptide Self-Assembly Assessment by SICM

Fmoc-FF hydrogelation is known to include several stages of self-assembly. Here, we performed a comparative analysis of optical microscopy (conventional fluorescence confocal microscopy and FLIM) and SICM in measuring the morphological changes accompanying the Fmoc-FF self-assembly in aqueous solution. Herein, the first two methods were applied in order to understand the results obtained by SICM technique and to interpret them rightly. To perform fluorescence imaging, the fluorescent probe Thioflavin T (ThT) was used [33,34,35].

During the hydrogelation process, the FLIM images were obtained and can be seen in Figure 1A.

The self-assembly and gelation process of Fmoc-FF was studied using the solvent-switch method [36]. In the process of gelation, firstly, the turbid Fmoc-FF solution was observed that became transparent with time, when the hydrogel is formed. FLIM measurements revealed that several stages could be identified upon Fmoc-FF self-assembly. Initially, the spherical structures appeared when the Fmoc-FF stock solution was diluted in water (Figure 1A, stage I). Then, these spheres increased in size (Figure 1A, stage II) and the transition from spherical aggregates into fibrils was observed (Figure 1A, stage III). The fibrillar gel was the final stage of hydrogel formation (Figure 1A, stage IV).

Next, structural changes during Fmoc-FF hydrogel formation in aqueous solution were investigated by SICM in real time (Figure 1B). Contrary to the FLIM measurements, the transition from spherical structures (stage I) to fibrillar hydrogel (stage IV) was not observed, and the process of hydrogelation was represented by fibril growth. This is due to the fact that SICM provides the image of peptide aggregates directly on the surface of the Petri dish, as the pipette passes through the gel and reaches the bottom, where it scans the formed fibrils. Hence, the hydrogelation as observed using SICM demonstrated 2D self-assembly of peptides on the surface: first, small peptide aggregates appeared on the surface of the Petri dish (Figure 1B, 0 min). Then, spherical particles precipitated from the volume of the sample onto the surface (Figure 1B, 16 and 32 min), and, finally, the densely packed layer of fibrillar aggregates was formed (Figure 1B, 40, 48 and 56 min). Hence, SICM observations are consistent with a growth mechanism consisting of nucleation, growth, and fibril formation of peptide aggregates during hydrogelation.

In these experiments, SICM was combined with fluorescence confocal microscopy (Figure 1C), which allowed scanning of the sample along the *z*-axis (Figure 1D). For the mature peptide-based hydrogel, aggregates were adsorbed on the surface (0 μm), while 10 μm above the surface the system was mainly characterized by fibrillar structures, see Figure 1C. This observation is in agreement with the SICM analysis, which showed non-fibrillar peptide aggregates adsorbed on the surface.

Thus, when studying peptide self-assembly in 3D, SICM reveals the processes near the surface as the system rigidity at the initial stages of hydrogelation is not enough to provide sufficient contrast for the pipette electrode. On the other hand, the obtained results show that SICM can be used for in situ imaging of self-assembly in 2D, making it a perfect tool for investigating biomolecules aggregation on the surface [37,38].

### 3.2. Topography Measurements of Cells in the Peptide-Based Hydrogel

Next, SICM was used for the in situ visualization of cells growing on the hydrogel (Figure 2). In addition, since certain concentrations of ThT have been shown to modulate the self-assembly pathway and modify the hydrogel morphology and mechanical properties [35], hydrogel formation in the presence of ThT was also analyzed. For this purpose, MCF-7 epithelial cells were seeded on three different substrates: (1) the control substrate (Petri dish, Figure 2A); (2) Fmoc-FF hydrogel + ThT (Figure 2B); and (3) the Fmoc-FF hydrogel (Figure 2C).

The topography of mature hydrogels surface as revealed by SICM is shown in Figure 2D,E. Both systems consisted of long thin fibrils, which were denser in the presence of ThT, in agreement with previous results [39]. We note that SICM allowed for fibrils imaging with better resolution than optical microscopy, for instance, a characteristic thickness of 200 nanometers was demonstrated for the fibrils in the Fmoc-FF hydrogel (Figure 2A).

Notably, Ushiki et al. previously obtained SICM images of individual collagen fibrils; however, to achieve this goal, the tendon of adult Wistar rats was placed on a glass surface and dried overnight, after that the sample was immersed in the physiological solution [39]. Herein (Figure 2), no sample drying was applied, as the hydrogel formation occurs in aqueous solution. To compare the effect of drying on the morphology of the Fmoc-FF hydrogel, the formed samples were dried overnight and then placed in a buffer solution and visualized by SICM. In this case, the topography mapping process was greatly simplified, and the fibrils of the hydrogel could be much more evidently observed (Appendix A in the Supporting Information) compared to commonly-used hydrogel that consists of fibrils that are thicker as they are “swelled” in aqueous solution. The average thickness of fibrils that were dried overnight was d = 0.33 μm, while for fibrils that were studied in aqueous solution this value was d = 0.43 μm.

SICM measurements revealed that the shape of the cells grown on the hydrogel significantly differed from the control substrate (Petri dish). Thus, the cells grown in the hydrogel were more spherical while the cells grown on the petri dish were more spread. The non-deformation effect of the hydrogel indicates that the hydrogel better simulates the physiological environment of the cells.

### 3.3. Characterization of the Hydrogel Using a Zwitterionic Label-Free pH Nanoprobe

The physical and chemical properties of hydrogels can strongly influence cell viability [40]. The maintenance of a relatively constant and neutral extracellular microenvironment is strongly required for cell survival and proliferation. Fmoc-FF gelation is mainly provided either by lowering the pH of the aqueous solution (pH-switch method) or by adding water to a peptide solution that was initially dissolved in organic solvent such as dimethyl sulfoxide, DMSO (solvent-switch method). The final pH of the solution is considered as the main factor for difference in mechanical properties for the hydrogels obtained by these different protocols [41]. Hence, not only the method of gel preparation influences its properties, but also the pH value of the system.

Herein, the preparation of the hydrogel was carried out by the solvent-switch method as gel formation occurs rather quickly, requiring 5–30 min (depending on the peptide concentration, temperature, etc.). The hydrogel structure was homogeneous in comparison with hydrogel obtained by the pH-switch method where the hydrogel was forming after 18 h and its structure was more liquid and might have contained peptide aggregates. The final pH in the hydrogel prepared by the solvent switch method was 4.8. The Fmoc-FF hydrogel was washed several times with buffer for several hours to set the pH to 7.3 which is suitable for cell growth. To investigate the final pH value in the hydrogel before cell adhesion, characterization of the hydrogel using a zwitterionic label-free pH nanoprobe was carried out.

The label-free pH-sensitive nanoprobe consists of a self-assembled zwitterion-like nanomembrane at the tip of a nanopipette that was fabricated using a mixture of glucose oxidase (GOx), containing negatively charged carboxylic acid residues and poly-l-lysine (PLL) as well as positively charged quaternary amines crosslinked with glutaraldehyde (Figure 3A). Such nanomembranes allow the ion current to flow through the membrane matrix and exhibit preferential permeability to anions at low pH and cations at high pH, making SICM an appropriate technique for hydrogel high-resolution pH mapping.

Upon varying the pH of the solution, the dependence of the current on the voltage demonstrated the ion-current rectification for the pH nanoprobe (Figure 3B(I)). Using this dependence, the calibration curve at a constant voltage of 600 mV was obtained (Figure 3B(II)), which was further used for the evaluation of the pH inside the hydrogel and outside (in the buffer) under different experimental conditions (Figure 3C). First, the pH for the Fmoc-FF hydrogel with an initial pH value of 4.8 was measured. Good sensitivity of the pH nanoprobe was demonstrated as the pH value inside the hydrogel did not change and showed a permanent pH value of 4.8. After verification of the pH nanoprobe sensitivity, the Fmoc-FF hydrogel that was characterized by pH 4.8 after its preparation was put into a buffer at pH 7.3 and left for 16 h. The measurements demonstrated that the pH of the hydrogel changed completely to the pH of the buffer and that no pH gradient with depth was observed (Figure 3C(II)).

Since physiological media of living cells are typically buffered solutions, the definition of the H+ gradient in hydrogel is of high importance for the characterization of scaffold properties. Herein, it is demonstrated that a pH-sensitive nanoprobe can be used for the precise mapping (with an accuracy of 0.01) of the hydrogel pH, making it possible to investigate 3D pH distribution in biomaterials in situ.

### 3.4. Assessment of ROS in Cells Grown on the Fmoc-FF Hydrogel

Next, electrochemical method using platinum electrode was used to assess the ROS in cells grown on hydrogels as a marker of the cellular response to microenvironment changes. In general, ROS include different types of oxygen radicals: the superoxide anions, hydroxyl radicals, hydrogen peroxide, nitric oxide, peroxynitrite and singlet oxygen [42]. Hydrogen peroxide (H_2_O_2_), that is freely diffusible and relatively long-lived, is usually considered as a main agent in stress signal transduction pathways [43]. Herein, H_2_O_2_ levels in MCF-7 grown on different substrates (Petri dish surface (control), Fmoc-FF hydrogel and Fmoc-FF hydrogel in the presence of ThT), were measured using electrochemical method using platinum electrode (Figure 4).

In this experiment, a platinum electrode filled with carbon was used as a nanosensor. The electrode penetration into the cell (Figure 4A) stimulated the current increase in the system due to the cell’s mechanical stimulus and double-electric layer disturbance, followed by its exponential decrease over time (Figure 4A(I)). Upon reaching the current equilibrium, the electrode was removed from the cell and the difference of current inside and outside of the cell was measured (ΔI).

Since bare carbon nanoelectrodes are mostly irresponsive to H_2_O_2_, the platinum deposition on the electrode significantly increases the sensibility of the nanoelectrode. The electrodes were calibrated in H_2_O_2_ solutions ranging from 0.1 to 100 μM (the information about electrode fabrication and calibration can be found in the SI). The electrode calibration curve for H_2_O_2_ is presented in Figure 4A(II).

The photos of cells in Figure 4B demonstrate that the shape and adhesion properties of MCF-7 are strongly defined by their environment: the cells in the hydrogel attached tightly and look more “physiologically”, whereas the cells on Petri dish are spread over the surface. It was observed that the current value, and therefore the intracellular H_2_O_2_ concentration was two-fold higher in MCF-7 cells grown on the control substrate compared to the Fmoc-FF and Fmoc-FF+ThT hydrogels, that indicates no effect of self-assembled peptide hydrogel on the functions of living cells.

## 4. Conclusions

The results of the present work show that SICM is extremely useful for the study of self-assembled peptide-based hydrogels under physiological conditions. The topography of the formed hydrogel and of living cells grown on it was analyzed. The shape of the cells grown on the hydrogel differed strongly from cells grown on the control substrate (Petri dish). The non-deformation effect of the hydrogel on the shape of the cells indicates that the hydrogel better recapitulates the physiological environment of the cells. A label-free pH-sensitive nanoprobe was used for the characterization of the hydrogel pH, demonstrating that the pH of the hydrogel was entirely determined by the buffer that covered its surface, making SICM an appropriate technique for hydrogel high-resolution pH mapping. Moreover, the SICM technique was applied to evaluate the cell growth and survival on the Fmoc-FF hydrogel by estimation of intracellular ROS concentrations, indicating functions of living cells growth on self-assembled peptide hydrogel.

## Figures and Tables

**Figure 1 cells-11-04137-f001:**
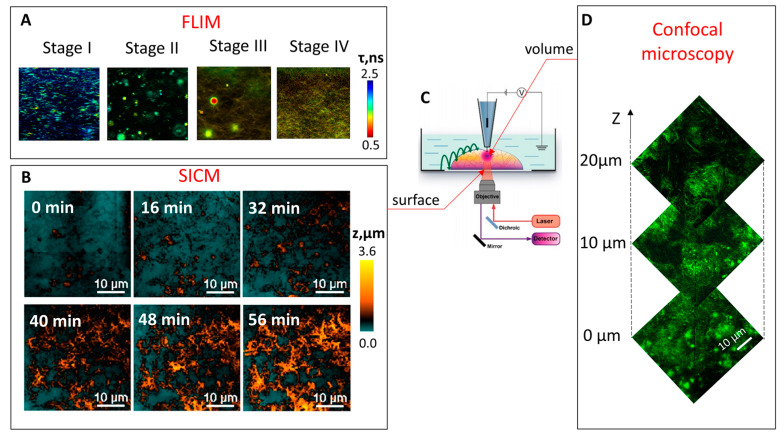
(**A**) FLIM images of the Fmoc-FF hydrogel formation at different stages in the presence of 40 µM ThT. Image size 80 × 80 µm^2^. (**B**) The process of Fmoc-FF self-assembly as revealed by SICM. (**C**) Schematic representation of areas in the sample from which by SICM and confocal microscopy images are obtained (**D**) Confocal microscopy Z-stack images of peptide Fmoc-FF aggregates.

**Figure 2 cells-11-04137-f002:**
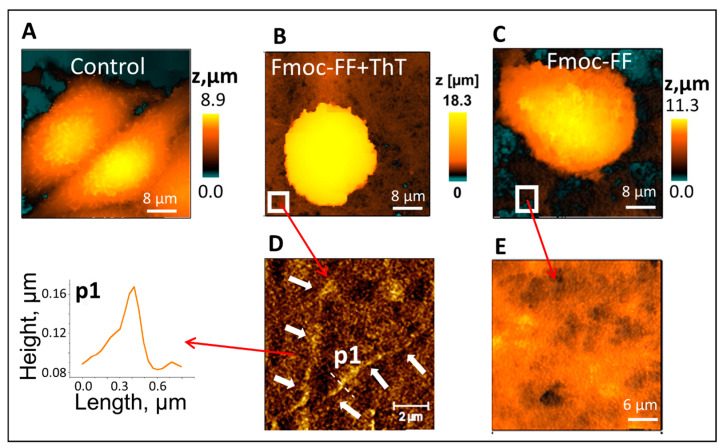
SICM topography imaging of the hydrogels and living cells grown on them. (**A**–**C**) The topography of living cells (MCF-7) grown on (**A**) control substrate, (**B**) Fmoc-FF+ThT hydrogel, (**C**) Fmoc-FF hydrogel. (**D**,**E**) The topography of (**D**) Fmoc-FF+ThT and (**E**) Fmoc-FF hydrogel. In the insert of figure (**D**), an image of a single fibril is shown. White arrows show single fibrils topography of the Fmoc-FF+ThT system. Concentrations: Fmoc-FF = 5 mg/mL, ThT = 40 µM.

**Figure 3 cells-11-04137-f003:**
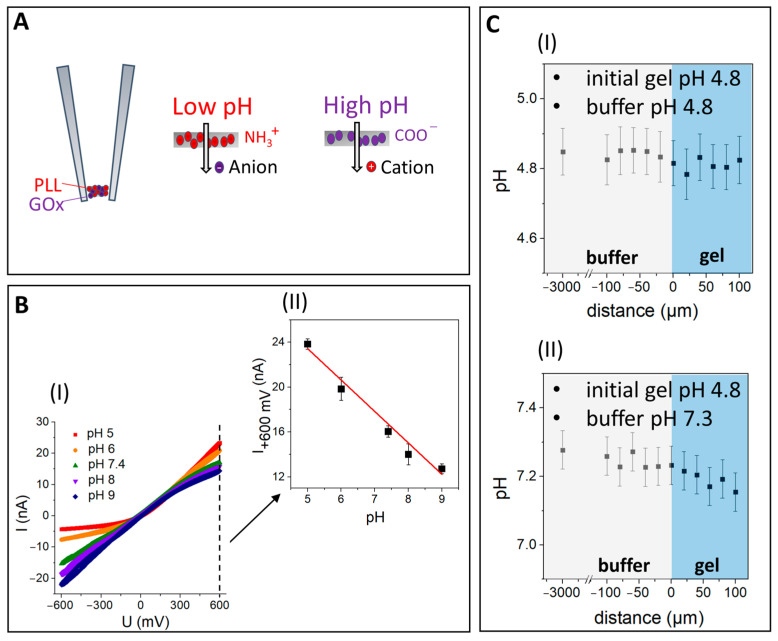
(**A**) The principle of operation for the sensor where the nanomembrane demonstrates preferential permeability for anions at low-pH and for cations at high-pH. (**B**(**I**)) Current-voltage characterization of the sensor at varying pH values. (**B**(**II**)) The dependence of the current on pH at 600 mV. (**C**) The pH dependence of media in the buffer (gray) and inside the hydrogel (blue) at a distance from hydrogel surface for (**I**) hydrogel with initial pH value of 4.8 that was filled with buffer (pH 4.8) and (**II**) hydrogel with initial pH value of 4.8 that was filled with buffer (pH 7.3).

**Figure 4 cells-11-04137-f004:**
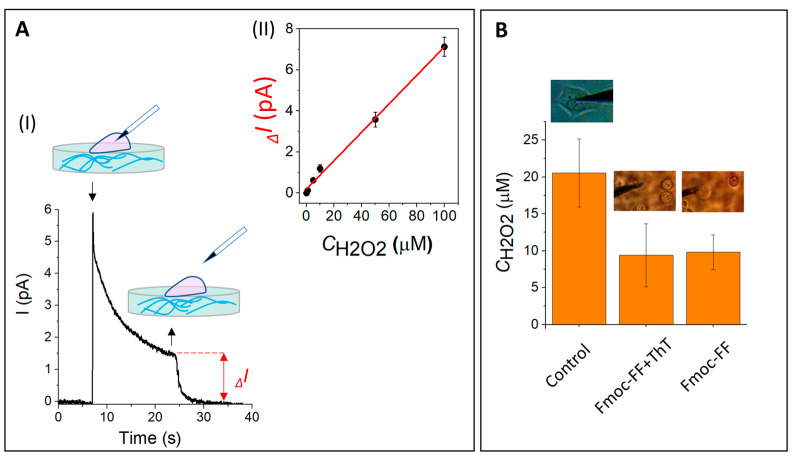
ROS assessment in living cells using the SICM technique. (**A**(**I**)) Current versus time plot inside and outside MCF-7 cells grown on a Petri dish (control). The “down” arrows indicate the moment of penetration inside the cells and the “up” arrows indicate the moment of retraction of the sensor from the cells, respectively. (**A**(**II**)) The electrode calibration curve for H_2_O_2_. (**B**) The H_2_O_2_ content for MCF-7 cells grown on the control substrate, Fmoc-FF + ThT hydrogel and Fmoc-FF hydrogel. Above the data the photos of electrode and the cells with their characteristic forms placed on control, Fmoc-FF + ThT and Fmoc-FF substates can be seen.

## Data Availability

Not applicable.

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
