# Peer review of "Sensing Cells-Peptide Hydrogel Interaction In Situ via Scanning Ion Conductance Microscopy"

_cells, 2022, doi:10.3390/cells11244137_

Round 1

Reviewer 1 Report

Tikhonova and co-workers have submitted a manuscript that addresses the question of Scanning ion conductance microscopy (SICM) possibilities for the investigation of cells-peptide hydrogel interaction in situ. The topic of the manuscript  fits the profile of the journal very well. Not only do the authors demonstrate conclusively such properties as ROS definition in living cells placed on the hydrogel and hydrogel pH characterization by this technique that allows to study cellular function under physical conditions, but also for the first time the way of hydrogel formation on 2D surfaces is performed that opens the perspectives of SICM applying for questions in 2D self-assembly processes at solid-liquid interfaces. Moreover, by SICM method the possibilities to investigate  living cells morphology on different substrates including hydrogel have been demonstrated.   The paper is well written and the data  is well presented. Therefore, I would like to recommend its publication after considering the following minor issues:

-In the "Materials" in section "Materials and Methods" it is written that the concentration % (v/v) of DMSO was 5% after dilution into water. Was it the final concentration of DMSO? Such solution may have influence of living cells functions. The procedure of hydrogel preparation needs to be stated clearer.

-In the section "Topography measurements of cells in the peptide-based hydrogel" it is written that hydrogel under physical conditions consists of thicker fibrils that are "swelled" in aqueous solutions in comparison with hydrogel that was dried overnight. I think it’s better to write about the characteristic sizes for these structures.

-It’s better to be consistent in the text when the authors describe the final structure of hydrogel – it’s better to use either "fibrils" or "fibers".

-What is shown in the images above the columns in the Figure 4B? It should be mentioned in the Figure caption.

-Please indicate the temperature for all measurements in the section "Materials and Methods".

Reviewer 2 Report

In the work of Tikhonova et al, an interesting study of the effect of a Fmoc-FF hydrogel scaffold on cell growth is presented. In particular, scanning ion conductance microscopy (SICM) is used to detect pH conditions at the interface.

In my opinion, the manuscript can be published with minor revisions.

My general comment is that the role of fluorescence and confocal microscopy combined to SICM in this work seems marginal. A better motivation for their combined use in this paper would be appreciated.

The use of carbon-filled micropipettes in Section 3.4 provides interesting information. I was wondering what is the difference of using commercially available carbon nanoprobes that you coat with platinum as described in the  supplementary materials, compared to carbon platinum-coated nanoprobes available from the same company.

The following list of minor points is intended to help the Authors to improve clarity of their manuscript.

261 tomography -> topography

315 The SEM micrograph in panel A is missing.
